# Four Novel SARS-CoV-2 Infected Feral American Mink (*Neovison Vison*) Among 60 Individuals Caught in the Wild

**DOI:** 10.3390/ani15111636

**Published:** 2025-06-02

**Authors:** Francesca Suita, Miguel Padilla-Blanco, Jordi Aguiló-Gisbert, Teresa Lorenzo-Bermejo, Beatriz Ballester, Jesús Cardells, Elisa Maiques, Vicente Rubio, Víctor Lizana, Consuelo Rubio-Guerri

**Affiliations:** 1Servicio de Análisis, Investigación, Gestión de Animales Silvestres (SAIGAS), Veterinary Faculty, Universidad Cardenal Herrera-CEU, CEU Universities, 46115 Valencia, Spain; francesca.suita3@uchceu.es (F.S.); jordi.aguilo@uchceu.es (J.A.-G.); jcardells@uchceu.es (J.C.); 2Department of Pharmacy, School of Health Sciences, Universidad Cardenal Herrera-CEU, CEU Universities, 46115 Valencia, Spain; miguelpadillablanco54@gmail.com; 3Viral Immunology Lab, Molecular Biomedicine Department, Margarita Salas Center for Biological Research (CIB-CSIC), 28040 Madrid, Spain; 4Department of Biomedical Sciences, School of Health Sciences, Universidad Cardenal Herrera-CEU, CEU Universities, 46115 Valencia, Spain; teresa.lorenzobermejo@uchceu.es (T.L.-B.); beatriz.ballesterllobell@uchceu.es (B.B.); emaiques@uchceu.es (E.M.); 5Instituto de Biomedicina de Valencia del Consejo Superior de Investigaciones Científicas (IBV-CSIC), 46010 Valencia, Spain; rubio@ibv.csic.es; 6Group 739, IBV-CSIC, Centre for Biomedical Network Research, Instituto de Salud Carlos III (CIBERER-ISCIII), 46010 Valencia, Spain

**Keywords:** American mink, animal reservoir, COVID-19, *Neovison vison*, One Health approach, SARS-CoV-2

## Abstract

The COVID-19 pandemic, caused by the SARS-CoV-2 virus, raised awareness of the role animals can play in the spread of infectious diseases. Mink are highly susceptible to this virus and can transmit it to other mink and humans. In this study, we tested 60 wild-living (feral) American mink captured in eastern Spain to detect possible SARS-CoV-2 infections. We found four positive animals. Combined with a previous pilot study that detected 2 infected mink among 13, only 6 out of 73 animals tested positive. This finding indicates that while infection can occur in the wild, the overall prevalence remains low. We also analyzed the body condition and reproductive status of the mink, which showed seasonal breeding and high population turnover, consistent with other wild mink populations. These results suggest limited spread of SARS-CoV-2 in feral mink and highlight the importance of continued monitoring of wildlife to detect emerging diseases that may affect both animals and humans.

## 1. Introduction

Pandemics have had a strong influence on human history [1]. The last pandemic, the first in the 21st century, was COVID-19. It was due to SARS-CoV-2, a coronavirus that appeared in late 2019, likely from a wild animal reservoir [2], and rapidly spread across the globe [3]. This pandemic, which resulted in unparalleled tolls on health, economic, and social life, reshaped public health strategies worldwide [4]. It provided very strong arguments for the One Health concept, which includes animals as well as the environment and the encroachments of humans into wildlands in disease emergence and the development of infectious threats to public health [5]. Thus, although human-to-human transmission was the major driver of SARS-CoV-2 spread, the detection of infection in animals after the virus had spread through the human population [6] raised strong concerns about the potential of non-monitored viral evolution in wild animal reservoirs [7,8].

Many domesticated animals (including cats, hamsters, ferrets, mink, raccoon dogs, and rabbits) proved capable of SARS-CoV-2 infections, including, in some cases, viral transmission to other animals [9]. Furthermore, wild animal species, whether living in captivity (largely great apes or zoo-held felines [10,11]) or in the wild (feral mink, otters, deer mice, woodrats, skunks, white-tailed deer), also proved vulnerable to SARS-CoV-2 infection [7,8,9,12]. To date, thirty-six animal species from sixteen taxonomic groups have been reported across forty-four countries as susceptible to SARS-CoV-2 under field conditions [5,13].

Among animals, mustelids occupy a prominent position, given their susceptibility to SARS-CoV-2 infection and ability to transmit the virus, to the point that [14] suggested the possibility of panzootic spread of SARS-CoV-2 deriving from infected mustelids. After humans and possibly white-tailed deer of North America [12], this is the taxonomic group with the largest number of SARS-CoV-2-infected individuals declared: ~26,000 reported in just eleven countries [15]; a large fraction of them (~9700 individuals) American mink (*Neovison vison*) from fur farms, with eighty-four outbreaks reported across seven countries [15]. In fur farms, mink COVID-19 infections initially derived from human caretakers, but then they were spread by animal-to-animal transmission, in some cases with transmission back to humans [6,9,16]. Infected mink often exhibit clinical signs such as weight loss, respiratory tract manifestations, and increased mortality [17]. Adaptive viral evolution appears fast in infected mink [9], and, in fact, the amino acid change mapping in the Receptor Binding Domain (RBD) of the S protein, S:Y453F, which increases the affinity for the ACE2 cellular receptor, emerged in a Danish mink farm, rapidly spreading through the human population [18,19].

Because of past escapes from farms, American mink has become an invasive, self-sustained feral species in Spain [16]. We [7] pioneered detection of SARS-CoV-2 in feral mink by finding two infected animals in a pilot study among thirteen trapped invasive feral American mink living in the wild in the high courses of two rivers of the Valencian Community (east of Spain, Mediterranean coast, Figure 1A). With the purpose of obtaining sounder indications on how frequent SARS-CoV-2 infection was in feral mink in these two river courses, we extended this study to sixty additional feral animals culled in these river courses in three campaigns taking place from November 2020 to May 2022. Here, we report findings of four additional SARS-CoV-2-positive animals, three from the Palancia river (one per campaign) and one trapped early in the Mijares river. Our data on these seventy-three animals, although reassuring given the low prevalence of infection, allowed us to make inferences from limited sequencing concerning viral evolution outside the eye of the health-monitoring system, highlighting the need for vigilance in these feral animals and the importance of the One Health approach.

Thus, in this article, we report the outcomes of this extended surveillance study, including prevalence data, virological characterization of positive cases, and ecological observations relevant to the zoonotic risk posed by feral mink.

## 2. Materials and Methods

### 2.1. Study Area, and Procurement of the Samples

As described in our prior pilot study on thirteen feral mink [7], the study area was the upper courses of the Mijares and Palancia rivers, two rivers of the Castellón province of the Valencian Community (east coast of Spain) (Figure 1A) that have separate courses from origin to end, emptying into the Mediterranean at coastal points separated by about 30 km. Their upper courses are separated by mountain ranges with peaks reaching 1000 m of altitude. These rivers and tributary ravines sustain feral populations of American mink resulting from farm escapes and deliberate releases at the end of the 20th century. The sixty American mink (thirty-two males and twenty-eight females) studied here were culled by wildlife rangers as part of an invasive species management plan. The dates of the three culling campaigns (mid-November 2020, mid-March 2021, and early May 2022) and the location of the trapping sites (UTM coordinates and municipal districts hosting each trapping site) are given in Table 1.

The treatment of the animals, including humane sacrifice, conservation, examination, necropsy, and collection of samples (presently nasal and rectal swabs, mediastinal lymph nodes, and lung tissue from all 60 animals), was reported in our previous works [7,8]. In short, at necropsy, each sample was taken and placed aseptically in a plastic tube containing 0.5 mL Sample Preservation Solution (ref. P042T0020100, JiangSu Mole Bioscience, Taizhou, China; sold in Spain by Palex Medical, Madrid, Spain). This proprietary commercial solution is used to inactivate the virus and to preserve the RNA. Animals 49, 52, and 54 (Table 1) were pregnant females. One fetus from each pregnant animal was taken randomly from among the several gestation sacs of the pregnancy (see below) for viral assay. The fetus was placed in its tube with 1.5 mL of Sample Preservation solution and was otherwise treated as the lung and mediastinal lymph node samples. The hermetically sealed tubes containing the samples in preservation solution were placed at −80 °C < 2 h after procurement, remaining at this temperature until their use in SARS-CoV-2 analyses.

### 2.2. Viral Detection and Molecular Analyses

Lymph node and lung tissue were homogenized as reported [7,8], and fetuses were processed in the same way. Total RNA was purified [7] using the NZY Total RNA Isolation kit (from NZYtech, Lisbon, Portugal), following the protocol outlined by the manufacturer. For viral detection, we carried out a custom-devised two-step RT-PCR procedure [7] as modified in [8]. Briefly, isolated RNA was first retrotranscribed to cDNA using the NZY First-Strand cDNA Synthesis kit (from NZYtech, Lisbon, Portugal), and then detection was based on a separate 40-cycle qPCR amplification carried out as reported [8] for the region of interest. Initial detection was based on qPCR amplification of nucleotides 28,701–28,951 of the viral genome (from here on, the numbering is that for the Wuhan-1 viral genome sequence, GenBank NC_045512.2) using primers 5’GCAGTCAAGCCTCTTCTCGT3′ and 5’TTGCTCTCAAGCTGGTTCAA3’. This amplicon corresponds to a highly conserved region of the nucleocapsid (*N*) gene of the virus [20]. Positivity was confirmed by separate qPCR amplifications of a region of the gene for the surface glycoprotein, *S* (bases 22,113–22,231; forward primer, 5′GGACCTTGAAGGAAAACAGG3′; reverse primer, 5’TGGCAAATCTACCAATGGTTC3’), or of the *ORF10* gene and flanking regions (bases 29,511–29,698; forward primer, 5′ATTGCAACAATCCATGAGCA3′; reverse primer, 5’GGCTCTTTCAAGTCCTCCCTA3’). qPCR procedures were performed exactly as in [8]. The correctness of these amplifications was supported by electrophoretic sizing of the qPCR product (illustrated in Figure 2) and by identification (Sanger sequencing) of the fragment produced in the amplification.

### 2.3. Sanger Sequencing and Phylogenetic Analyses

Following RT-PCR amplifications, the amplified DNA fragments were Sanger sequenced by a core sequencing service (Genomic Department, Centro de Investigación Príncipe Felipe, Valencia, Spain) using an automated ABI Prism 3730 instrument (Applied Biosystems, Foster City, CA, USA) and with the forward amplification primer serving as the sequencing primer. To find related SARS-CoV-2 sequences stored in the GenBank database, all sequences were run with BLASTN. The nucleotide and encoded amino acid sequences were aligned with the consensus SARS-CoV-2 sequence (GenBank NC_045512.2) using BioEdit ver. 7.2.5 software, which was also utilized for analysis and for determining the degree of identity of the recovered sequences [8]. For phylogenetic analyses, distance matrices were calculated, and tree topology was inferred by the maximum likelihood method based on p-distances (bootstrap on 2000 replicates, generated with a random seed) using the MEGA11.0 software [21].

## 3. Results and Discussion

### 3.1. Animals Trapped

Sixty dark brown American mink were caught in three culling campaigns for controlling the invasive feral population of these foreign mustelids, the first in mid-November 2020, the second in March 2021, and the third one in early May 2022. Eight animals were trapped in the Mijares river, of which four/one/three were trapped in the 1st/2nd/3rd campaigns, respectively (Table 1). The other fifty-two animals were trapped in the Palancia river course, of which twenty-nine, fourteen, and nine were trapped in the 1st, 2nd, and 3rd campaigns, respectively (Table 1). In our previous pilot study [7] on thirteen animals from both rivers, five animals (three and two from the Mijares and Palancia rivers, respectively) had been trapped in the last two months of 2020, and eight animals (three and five from the Mijares and Palancia rivers, respectively) in January 2021. Therefore, adding together the pilot and present studies, seventy-three feral mink have been tested for SARS-CoV-2 in these two rivers (Figure 1), of which forty-six represented a relatively early period of the COVID-19 pandemic, encompassing the end of 2020 and the first month of 2021.

### 3.2. The Body Weights of the Animals Provide Insight into the Local Population of Feral Mink

American mink represent a long-lasting invasive endemism in the Mijares and Palancia river courses, but there are no detailed studies about this feral population. Although tangential to this study, the body weights of the present 60 animals were recorded (Table 1) and provided some information on this population. Characteristically [22], these weights were larger for males (*n* = 32) than for females (*n* = 28) (Figure 3A; male median and mean weights ~350 g higher than for females). Top weights of males and females were < 1500 g and < 1200 g, respectively (Figure 3A), typical for wild mink but much lower than those for presently farmed mink (see for example [22]), as expected for a feral, self-sustained community.

Interestingly, the mean weight of males (Figure 3B) increased in each culling campaign relative to the previous one. Although only three males were culled in the third campaign, preventing statistical significance of differences for weights in that campaign relative to the prior ones, the mean weight was the highest for all groups, with little dispersion (1359 ± 82 g). In contrast, the mean weight for females appeared not to increase in the three culling campaigns (Figure 3B). To try to understand why, we analyzed the frequency distribution by weight of the animals (Figure 3C–F). Females trapped in mid-November 2020 (16 individuals) showed a trimodal pattern of weight frequency distribution (Figure 3C). The three Gaussian bells, from lower to higher weights (respective means ± SD, 616 ± 53 g, 780 ± 31 g, and 1150 ± 21 g), comprised 61%, 27%, and 12% of the trapped females. The component of lowest weight and major abundance in terms of number of animals of this trimodal profile most likely represents immature juveniles that were born in the same year, being culled at an age at which young minks set out to find their own territories (https://www.havahart.com/minks-facts, accessed on 1 March 2025). This explanation is supported by the shift of this first component towards higher body weights in the nine females trapped in May 2022 (3rd campaign, Figure 3D), a month in which essentially all the surviving newborns from the previous year must have reached maturity. Actually, if the three pregnant females trapped in May are removed (since pregnancy should transiently increase weight in part due to the multiple gestation sacs with fetuses, see Figure 3G), the frequency distribution fits a broad Gaussian bell (instead of the bimodal distribution shown) with a mean value of 790 ± 110 g (curve not shown), a mean value that is very similar to the mean weight of the second Gaussian bell for the animals trapped in mid-November of 2020. This shift towards higher weights was also observed in the male cohorts when examining their frequency distribution for weight in the first and second culling campaigns, with 65% of the animals weighing <1 kg in mid-November 2020, while in the March campaign only 18% of the trapped males were <1 kg. All this suggests that the majority of animals in the population were born the same year, in line with the conclusion of a Danish study [22] that established that population turnover for American mink living in the wild was very fast because of low long-term survival rates.

The main conclusion of these observations is that the feral population in this eastern Spanish location shares the life cycle characterized in other locations in Europe or North America [22] (https://www.havahart.com/minks-facts; accessed on 1 March 2025), in which new kits are born in late spring or early summer, they search for their individual territory in the fall after they are born, and they become mature at ≥12 months. Further support for this conclusion stems from the fact that pregnant females (animals 49, 52, and 54; Table 1) were only found in the May culling, in line with the established time of delivery for this species in late spring or early summer. Pregnant females (Figure 3A; blue dots) fall at the higher end of the female weight distribution, likely due, as already mentioned, to the gestational contents and to physiological effects of pregnancy.

### 3.3. Four SARS-CoV-2-Positive Animals Were Detected Among the Sixty Animals Trapped

The major aim of our study was to assess the infection status of the trapped animals concerning SARS-CoV-2. With this goal, we carried out qPCR studies on the cDNA obtained by retrotranscription of total RNA isolated from four samples (nasal and rectal swabs and lung and mediastinal lymph node tissues) obtained from each one of the 60 animals. As already indicated in the Section 2, and with the same purpose of SARS-CoV-2 detection, we also obtained RNA from a fetus taken randomly from each one of the three pregnancies found at necropsy in female animals 49, 52, and 54 (Table 1). The RNA was immediately retrotranscribed to cDNA, storing it at −20 °C. Then, when all the cDNAs had been produced, we carried out parallel qPCR assays for the N gene in one type of sample at a time, using SYBR green. Of the 243 samples submitted to *N* gene-focused qPCR, only 6 samples belonging to four animals (animals 16, 29, 40, and 56) yielded a positive result (Table 2). The positivity was not based only on the fluorescent output in the qPCR assay but also relied on the observation by agarose gel electrophoresis of a product of the expected size (illustrative example in Figure 2, top left panel) and on subsequent Sanger sequencing confirmation of the correctness of the site that was amplified. This result was confirmed for the six positive samples by repeating the RNA extraction, retrotranscription, and *N*-gene-focused analysis of the newly prepared cDNA sample. Then, the cDNAs that were positive for the *N* gene, in parallel with a random selection of a few *N* gene-negative samples, were subjected to another two qPCRs focused on the *S* and *ORF10* viral genes. As expected for the presence of the entire viral genome in the *N* gene-positive samples, these same samples were found to be positive for the *S* and *ORF10* genes, using the same criteria as for the *N* gene. These criteria are the positivity in the qPCR assay, the observation of a product of appropriate electrophoretic size (Figure 2), and a Sanger sequence of the product that confirmed the identity of the amplified region.

In summary (Table 2), from the 60 animals tested, only the already mentioned four (animals 16, 29, 40, and 56) were positive, two of them (animals 16 and 40) exhibiting positivity for the nasal swab samples, and the other two (animals 29 and 56) for the rectal swab samples, with no animal yielding positivity for both swabs. Only in animal 29 the lung and mediastinal lymph node also were positive. This was the only animal that was positive for any of these internal tissues. The pregnant females were SARS-CoV-2-negative, and the three fetuses tested (one per dam) were negative, too. The two animals (animals 16 and 40) for which the nasal swab was positive, exhibited qPCR positivity with Ct values ≥ 29 for all three genetic regions examined, suggesting low viral loads. Animal 29, the one for which lung and mediastinal lymph node were positive, presented the lowest Ct values among the four SARS-CoV-2 2-positive animals, suggesting that it presented the highest viral load. Even in this case, the Ct values were >20 in the qPCR assays for the three genes in the three positive swab/tissue samples, supporting that the viral load was not very high.

Overall, the results are indicative of low prevalence of the virus among these 60 feral animals and suggest low viral load in those animals that hosted the virus. This aligns with the results of our prior pilot study [7]. The necropsies of the animals, which were agnostic concerning viral infection, did not record particular lesions in the animals proven later on to be infected, suggesting that the infections were subclinical. Furthermore, the body weight of the infected animals did not stand out as particularly low (Figure 3) relative to the bulk of the animals of the same sex.

While the total number of males and females in the 60-animal cohort was similar (28/32 females/males), three of the four infected animals identified presently were females. After pooling the results of the present study with those of our prior pilot study (total *n* = 73; 33 females and 40 males; from now on we will refer in this section to the 73 pooled pilot and present studies), the proportion of infected females remained higher than that of infected males (12.1% and 5.0%, respectively). This difference was not statistically significant (Fisher’s exact test), although it could be real if it reflects a higher exposition to contagion of the females than of the males because of extra food-searching exploration needed to satisfy the nutritional demand of the immature kits.

The distribution of the positive animals in terms of time of culling closely parallels the total number of animals trapped, since in the period defined by the first two campaigns of the present study and of the pilot study (end of 2020 and beginning of 2021), 61 animals were trapped and 5 were found to be infected (8.2%), while in the third campaign, which took place in 2022, only 12 animals were trapped and 1 animal (8.3%) was found infected. The proportion of the trapped animals that were found to be infected was 6.8% for the Palancia river and 14.3% for the Mijares river, two values that were not statistically different (Fisher’s exact test). When focusing on the animals trapped in the Palancia river, three of the four infected animals found in that river were trapped in the last 10 km of its high course, in the counties of the municipalities of Segorbe and Soneja (Figure 1). This concentration of positive animals grossly parallels the number of mink trapped in these sites, which accounted for 49% of all the trappings made in this river. The same can be said for the Mijares river, where the sites of trapping of the two positive animals were localized within 8 km of the high course of the river, the same area that concentrated 64% of all the mink trapped in this river.

### 3.4. Partial Gene Sequencing Reveals Common Traits with the B.1.177 and Alpha Variants of SARS-CoV-2

Although we used short stretches of Sanger sequencing just for confirmation of the region that had been amplified with each primer pair (see Materials and Methods), thus using only unidirectional sequencing utilizing a single sequencing primer (the forward one of the amplification) per amplicon, the quality of the sequences for a large part of the three amplicons was high enough for identification of sequence variants. This was the case (Figure 4) for nucleotides 22,150–22,252 of the *S* gene (encompassing from the last base of codon 196 to the last base of codon 223 of the *S* gene coding sequence); nucleotides 28,734–28,951 for the *N* gene (encompassing from the 2nd base of codon 154 to the last nucleotide of codon 226 of the *N* gene coding sequence); and nucleotides 29,572–29,698 in the *ORF10* gene region (encompassing from the second base of codon 5 until the last codon of this gene coding sequence, plus 24 downstream flanking bases of non-coding sequence; Figure 4). Although these regions were quite short, they revealed some differential sequence traits of interest. The sequences were identical (summarized in Table 2) for the three positive samples (rectal swab, lung tissue, and a mediastinal lymph node) of animal 29, supporting viral variant homogeneity within the same animal, not favoring the possibility of prolonged infection with the opportunity for separate local evolution of the virus in different tissular foci. 

The partial sequences of the *S* and *ORF10* genes were identical in all four animals, while the *N* gene partial sequence differed in two animals (animals 16 and 40) from that in the other two positive animals (animals 29 and 56) (Table 2 and Figure 4 and Figure 5). The *S* partial sequence was the shortest among the three regions sequenced (just 82 nucleotides), but it included codon 222, a site of early variation in the SARS-CoV-2 pandemic, where the amino acid change (relative to the Wuhan-1 sequence) A222V emerged in Spain and spread through Europe in the summer of 2020 [23]. Our four positive animals had identical partial *S* gene sequences, which were identical, too, to the Wuhan-1 sequence, and thus, they did not encode the A222V change in the S protein. This change appears to favor subtle changes in the dynamic behavior of the receptor-binding domain (RBD) of homotrimeric S protein in the human receptor-binding-competent “up” conformation [24]. Phylogenetic evidence strongly suggests that this change has emerged independently on several occasions and in different genetic backgrounds throughout the SARS-CoV-2 pandemic [24], suggesting that it may enhance infectivity in humans, facilitating its establishment. This might not be the case for mink, in view of the absence of this change in our four positive feral animals, and also since among approximately 700 SARS-CoV-2 sequences obtained in Denmark from farmed mink from June till to November-–December of 2020, this change was only found in one infected animal [25].

Unlike the partial sequence of the *S* gene, the partial sequences of the *ORF10* and *N* genes found in our four positive animals showed differences relative to the canonical Wuhan-1 sequence. The variation in the *ORF10* gene, identical for our four animals, was the base change 29645G > T (coding sequence, *ORF10*:88 G > T), causative of the amino acid substitution ORF10:V30L. The isolates from animals 16 and 40 presented in the *N* gene partial sequence the non-synonymous substitution 28932C > T (coding sequence *N*:659C > T) causative of the amino acid substitution N:A220V, whereas the isolates from animals 16 and 40 presented in this same gene the replacement 28881–28883GGG > AAC (*N* coding sequence, *N*:608–610GGG > AAC), encoding the double amino acid substitution in adjacent residues of the N protein, R203K/G204R. All these changes have been reported previously in viral isolates from humans. The amino acid changes N:A220V and ORF10:V30L, present in the viruses from animals 16 and 40, are characteristic of the B.1.177 viral variant, which was of high incidence in Spain during the end of 2020 and the winter of 2021 [23]. In agreement with this, phylogenetic analyses show (Figure 5B,C) that our partial sequences for the *N* and *ORF10* genes from the isolates of animals 16 and 40 cluster with the B.1.177 variant. However, these viral isolates lack the A222V substitution in the S protein that also characterizes the B.1.177 variant [23]. This is reflected in the close clustering of the partial *S* gene sequences of our animals with the *S* gene of the reference Wuhan-1 genome (Figure 5A) rather than with the *S* gene sequences of B.1.177 or of other viral variants. These *S* sequences also clustered with those from farmed mink in the Netherlands (GenBank Accession Numbers OM758316.1), Denmark [19,25], and Italy (MT919525.1; [26]), which would be expected if the S:A222V replacement were not fit for SARS-CoV-2 infection in mink, possibly related to differences in the ACE2 receptors of mink and humans. Animals 16 and 40 were trapped in geographically close places in the same river (Palancia), and thus it would not be surprising if they were infected with the same viral variant, which might have persisted in these locations at the two times of capture (mid-November 2020 and mid-March 2021).

The three-nucleotide replacement 28881–28883GGG > AAC identified in the *N* gene of our positive animals 29 and 56 is typically associated with the Alpha (B.1.1.7) variant of SARS-CoV-2 [27], as shown by the clustering with this variant of the isolates from these animals in the phylogenetic tree of the *N* gene partial sequence (Figure 5B). The Alpha variant surfaced in the UK in the late summer or early fall of 2020 [28], some time earlier than animal 29 was trapped, whereas animal 56 was trapped in May 2022, two months later than the declaration by the World Health Organization that the Alpha variant was extinct among humans. Thus, the origin and persistence of this isolated sequence trait among those traits typical of the Alpha variant in this relatively remote population of feral mink deserves further study. This need is supported, too, by the fact that animals 29 and 56 were trapped not only at dates separated in time by 18 months but also in independent river courses and in locations that are quite apart from those in which the viral isolates of the B.1.177 lineage were identified in the other two mink found positive in the present study.

### 3.5. Final Considerations

The present data, revealing modest numbers of infected individuals in a relatively large cohort of animals culled in periods of high COVID-19 prevalence in the human population, are reassuring concerning the infectability in the wild of this highly susceptible species to the SARS-CoV-2 virus. This information is important, given the existing concerns about the spread of the virus among animals and potential risks to wildlife [29], particularly since infected mustelids have been pointed out [14] as having the potential for panzootic spread of SARS-CoV-2. If this potential is real, our study makes it difficult to attribute such potential to American mink living in the wild, possibly because of their living habits of low sociability and strict territoriality. Other traits of American mink, such as their life cycle, seasonality of mating, age to maturity, and high turnover in the wild, also appear to be retained by the feral invasive animals living in our temperate Mediterranean area, judged from our observations on the weights of the animals in this cohort.

In summary, the extensive (for a feral population) study carried out now supports the conclusion of our previous pilot study [7] that feral mink living in the wild can become infected, but it strongly suggests limited infection rates in the wild. This is in stark contrast with the situation encountered in mink farms [17], where mink-to-mink transmission is highly efficient, facilitated by the closeness to humans and by the physical proximity between animals that is typical in farm settings. Furthermore, it also differs from observations with more social animals living in the wild, such as white-tailed deer in North America. Feng et al. [12] reported evidence suggesting enzootic transmission within the deer population, allowing the virus to persist over time, while such intraspecies transmission may be more limited in feral or wild mink, which are less likely to encounter other mink or humans than deer living near urban or suburban areas.

We previously hypothesized for mink [7] and for wild otter [8] that SARS-CoV-2-contaminated waters might be the original source of the virus in those animals that are infected. SARS-CoV-2 has been shown to persist for some time in water [15] and to be present in human stools [30] and in wastewater, where it appears to derive largely from human excreta [15,31,32]. Specifically, in the province where our mink live (Castellón), Barberá-Riera et al. [31] detected SARS-CoV-2 in wastewater and recovered the genome sequence of the dominant viral variant in humans at the time of their study (B.1.177), whose sequence is similar to the one in some of our trapped feral mink, as shown by our phylogenetic analyses. Nevertheless, the low frequency of infection among our animals suggests that the waters of these two rivers do not sustain high transmission rates.

The distribution of positive samples across different anatomical sites in our study raises important questions about the stage and nature of SARS-CoV-2 infections in feral mink. In individuals 16 and 40, the only positive sample was the nasal swab, which may indicate early stages of infection. During this phase, viral loads tend to be concentrated in the upper respiratory tract, as the virus primarily targets cells in the nasopharynx or in the nasal mucosa [33]. Nasal swabs have been demonstrated to reliably detect high viral loads during this initial phase, before the virus disseminates to other parts of the body [34,35]. In contrast, individual 29 tested positive in the lung, mediastinal lymph node, and rectal swab, suggesting a more advanced stage of infection, potentially reflecting systemic viremia. The detection of the virus in multiple tissue sites implies viral spread beyond the initial entry point, possibly through the tracheobronchial tree or systemic circulation [33,36]. Finally, individual 56 tested positive only in the rectal swab, which may suggest a late infectious phase characterized by fecal shedding of viral RNA. This pattern aligns with observations that fecal shedding of SARS-CoV-2 RNA can persist for extended periods, often after respiratory swabs test negative [30,31].

Our study is not exempt from limitations. The more important one concerns our inability to draw further conclusions about potential variants due to our limited sequencing approach. The fact that we could not carry out next-generation sequencing (NGS) on our samples was largely due to the high Ct values, which were on average above 30, which we equated with a low viral load, while NGS typically requires a high viral load for reliable sequencing. Despite this limitation, our phylogenetic analysis provided valuable insights concerning variants, while the combination of the observed mutations and low viral load supports our suggestion that the life cycle of SARS-CoV-2 transmission among feral mink is not of major concern at this time. Nevertheless, this conclusion should not result in halting the monitoring and the genetic sequencing, even in the limited dimension performed in this study, since these measures are essential to assess potential wildlife adaptation of the virus that could result in novel zoonotic risks in case of re-entry of new variants into the human population.

## 4. Conclusions

This study expands our understanding of SARS-CoV-2 circulation in free-living feral American mink and demonstrates that, although infection is possible, its prevalence in the wild appears to be low. The detection of viral variants previously circulating in humans, along with the absence of clinical signs and the lack of evidence for vertical transmission, suggests that feral mink may not play a significant role in maintaining or spreading the virus under natural conditions.

Nevertheless, the presence of viral RNA in individuals sampled across time and space highlights the potential for sporadic spillover or limited intraspecies transmission. These findings reinforce the need for continued wildlife surveillance and genetic monitoring to detect changes in virus behavior or adaptation in animal reservoirs. As part of a broader One Health strategy, such efforts are essential for early warning and risk assessment of emerging zoonotic threats at the human–animal–environment interface.

## Figures and Tables

**Figure 1 animals-15-01636-f001:**
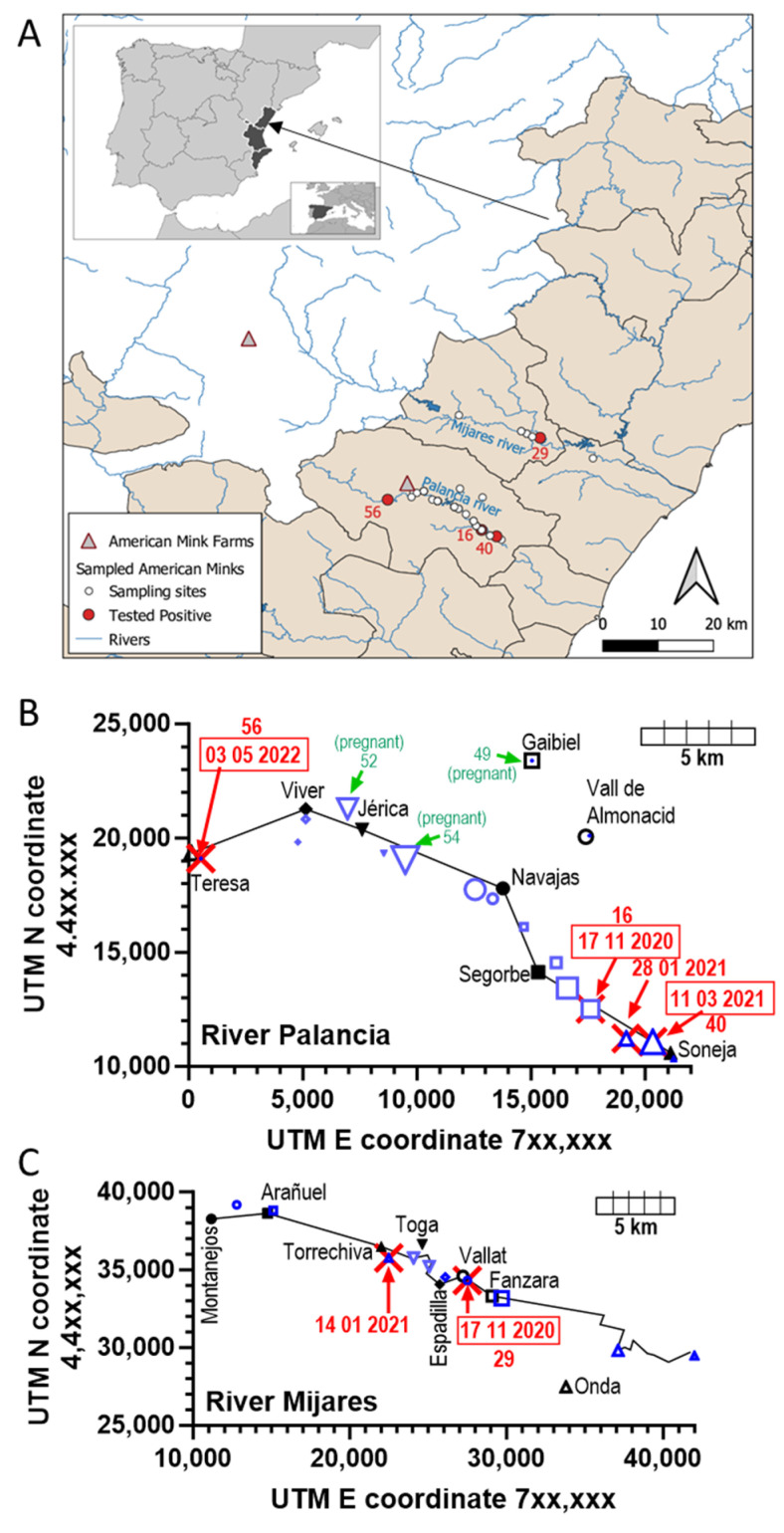
Study area and places where the animals were trapped. (**A**) Places where the present 60 animals were trapped are shown as white-filled circles except when a SARS-CoV-2-positive animal (identified by its number, referring to the leftmost column of Table 1) was trapped, in which case the circle is filled in red. Rivers are shown in blue, and the locations of the closest farms of American mink are indicated with triangles (red-lined and gray-filled). The inset shows the Valencian Community (in black) within the Iberian Peninsula. An arrow from the main panel shows what part of the Valencian Community is shown in this panel. The small inset within the inset shows the Iberian Peninsula (in black) in the Mediterranean. (**B**,**C**) Places (in blue) in which animals were trapped and analyzed in the present study in the upper courses of the Palancia (**B**) and Mijares (**C**) rivers. The data given are for 73 animals, resulting from the pooling of the present 60 animals with the 13 animals tested in our previous pilot study (see the main text). The sites are located according to their UTM coordinates (Northern Hemisphere; Earth’s surface section 30). For orientation, the municipalities to which the trapping sites belong are also located (black symbols and names), giving the same geometric shape to the municipality and its trapping sites. The size of the blue symbols increases with the number of animals trapped at a given site. In (**B**) this value goes from 1 to 8, and symbols have been represented in sizes 1 pt to 8 pt of GraphPad Prism 10, while in (**C**) the smaller symbols represent one trapped animal and the larger one two trapped animals. Sites where one animal was found to carry the SARS-CoV-2 virus are marked with red sword crosses, giving the date of trapping (format, Day Month Year; encased in rectangles for the present report, giving outside the box the number of the animal). Each site where a pregnant female was found is marked in green, giving the animal number (Table 1) and the word “pregnant”. The black lines in both panels link urban nuclei along the corresponding river, which in both cases flows from left to right. Thus, each one symbolizes a river course. The part of the Mijares river (**C**) downstream of Fanzara is plotted according to the river coordinates, to show the distance of the river course from the center of the city of Onda (~25,000 inhabitants).

**Figure 2 animals-15-01636-f002:**
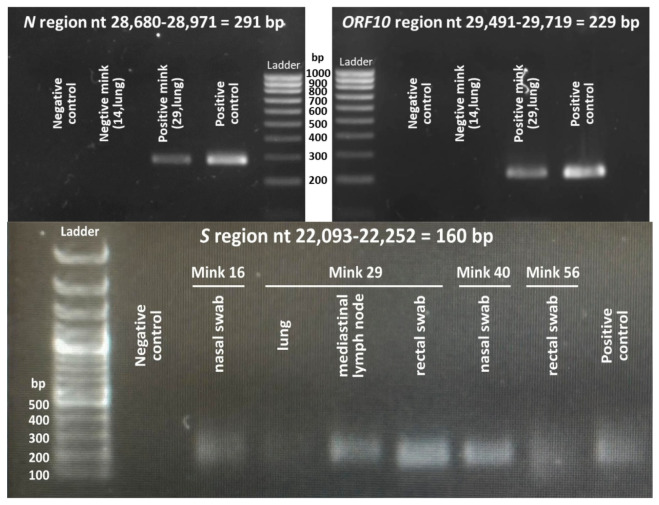
Illustrative agarose gel electrophoresis (top, 1% agarose; bottom, 2% agarose) of amplified products obtained by qPCR using for the indicated regions the primers given in Section 2.2 and, as templates, the cDNA retrotranscribed from the RNA isolated from the indicated samples. The positive control is a well-characterized retrotranscribed sample from a viral isolate of a human patient (not described here). The negative control used water as a template.

**Figure 3 animals-15-01636-f003:**
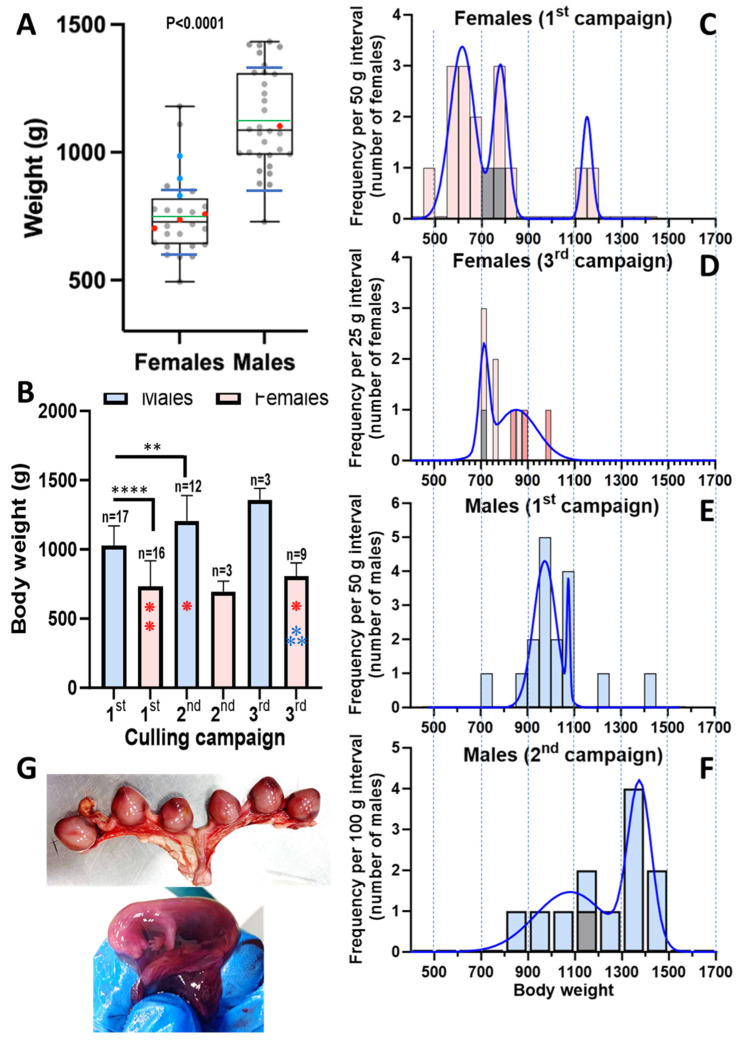
Body weights of the 60 feral mink studied here and images of a pregnancy in the May campaign. (**A**) Box and whiskers representation of the distribution of body weights among the 28 females and 32 males studied. The black and green lines that cross the box are, respectively, the median (females, 727 g; males, 1086 g) and mean (females, 753 ± 167 g; males, 1125 ± 190 g; the standard deviations are not shown in this figure). The bottom and top edges of the box define the first (Q1) and third (Q3) quartiles, and the whiskers give the total span of the measurements. The space between the blue transversal lines that cross the whiskers corresponds to 1.5 times the range between Q1 and Q3 (interquartile range, IQR; for females, 642.5–808.5 = 166 g; for males, 991–1308 = 317 g) centered around the median. The dots represent individual animals, being red for SARS-CoV-2-positive animals and blue for pregnant females. The *p*-value shown indicates a significant difference between the mean body weight for females and males with the probability level indicated (Student’s t-test). (**B**) Mean ± SD for body weight according to the sex and trapping campaigns. The number of animals for each bar is shown above it. Note that two groups had only three animals each. The **** and ** above the horizontal lines over the bars correspond to significant differences (Student’s *t*-test) between the corresponding means to respective *p*-values of <0.0001 and 0.0063. The red and blue asterisks denote the number of, respectively, SARS-CoV-2-positive animals and pregnant females (diagnosed at necropsy) in the corresponding groups. (**C**–**F**) Plots of frequency distributions of the most populated groups defined by sex and campaign. The class intervals are defined in the labeling of the axes. Note the fitting to bimodal and even trimodal (**C**) distributions. The blue lines are the fitting of the data to Gaussian or sum of Gaussian bells (GraphPad Plus 10 software). In the bars in which one animal was SARS-CoV-2 positive, this individual is colored gray. In addition, pregnant females are indicated (**D**) in the darker reddish color of their corresponding bars. (**G**) Pregnancy with six gestation sacs obtained at necropsy from the uterus of one of the three pregnant females. The inset shows detail of one of these sacs, opened to show the fetus, held in the gloved hand of the pathologist.

**Figure 4 animals-15-01636-f004:**
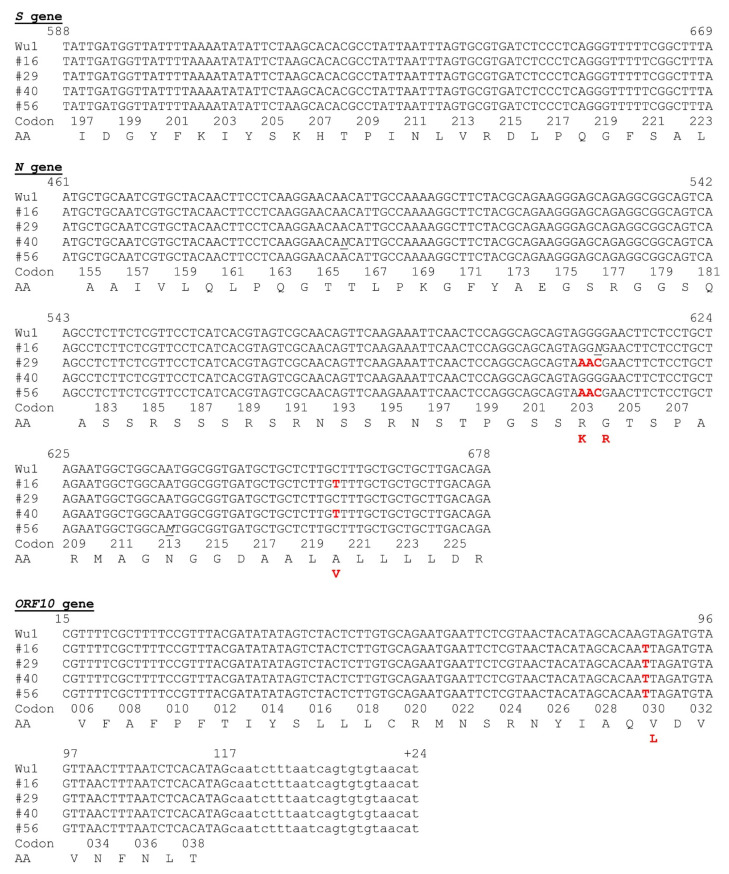
Alignment of the sequences obtained from SARS-CoV-2-positive mink for the regions amplified in the qPCR reactions for the *S*, *N*, and *ORF10* gene regions. Wu1 refers to the consensus Wuhan-1 sequence (GenBank NC_045512.2). Nucleotide numbering (above the sequence) is that for the Wuhan-1 consensus genome. The four sequences aligned with the Wuhan-1 genomic sequences are those for the positive animals (identified by the animal numbering as in Table 1). In animal 29, only one sequence is given per gene despite our sequencing of cDNA obtained from swab, lymph node, and lung tissue, since these viral sequences were identical. In the case of *ORF10*, the non-coding 3’ flanking sequence determined is shown in lowercase and is numbered preceded by a + sign, also giving the number of the last base of the coding sequence. In the sequencing of the *N* gene, there were a few bases showing ambiguity in the sequence. In these cases, an M denotes A or C, and a N indicates the possibility of any base at that position. Sequence replacements with respect to the reference sequence are shown in red. Codon numbers are given below the aligned sequences; for clarity, odd or even codon numbers are omitted. Amino acid residues are shown below the codon numbers in single-letter notation. Amino acids in red below the corresponding amino acid encoded by the Wuhan-1 consensus sequence (in black) give the amino acid substitutions encoded by the aligned sequences, with bases in red at the encoding codons.

**Figure 5 animals-15-01636-f005:**
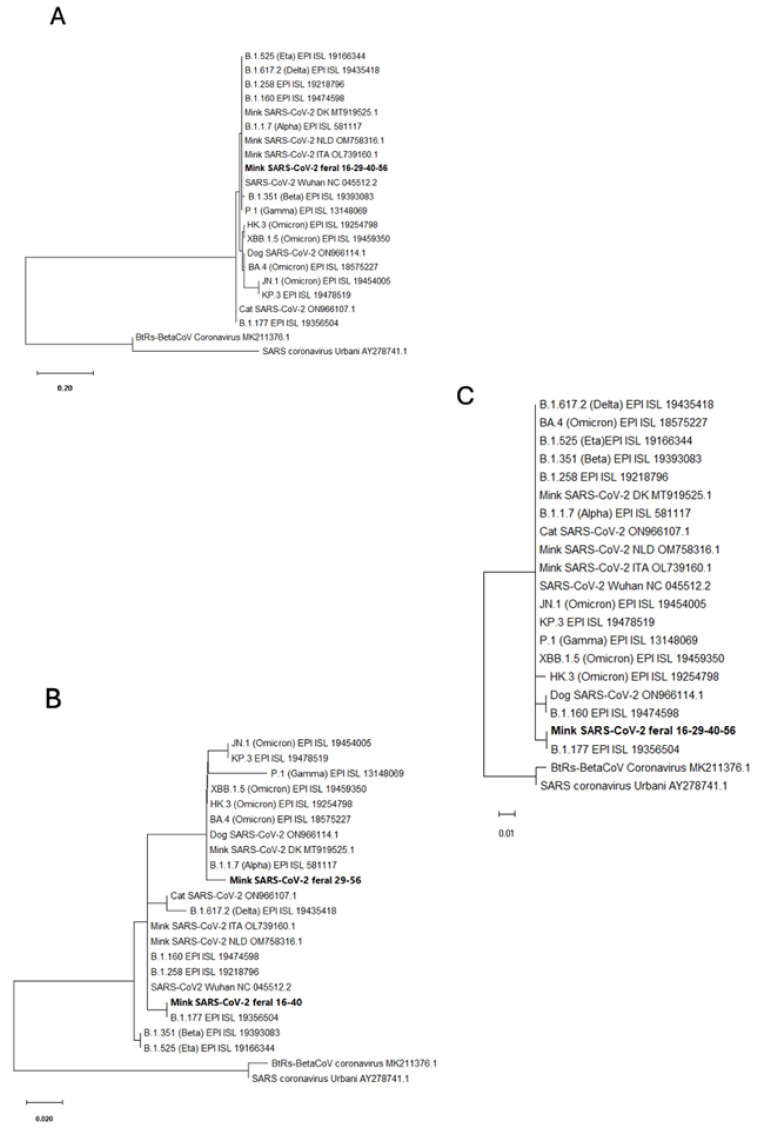
Molecular phylogenetic analyses based on partial gene sequences. These analyses used the regions defined by the consensus genome coordinates (GenBank accession number NC_045512.2) for nucleotides (**A**) 22,160–22,239 (corresponding to partial *S* gene sequence); (**B**) 28,871–28,964 (partial *N* gene sequence); and (**C**) 29,556–29,704 (partial *ORF10* gene sequence). Our mink SARS-CoV-2 sequences are highlighted in bold type, including at the end the appropriate GenBank Accession Numbers. Other sequences aligned have either Gisaid identifiers (EPI sequences) or GenBank identifiers (the last group of characters in the row, when they have two initial capital letters followed by a six-digit number, in some cases followed by 0.1 or 0.2). DK, NLD, and ITA correspond to Denmark, Netherlands, and Italy. The trees are drawn to scale, with branch lengths measured according to the number of substitutions per site (see scale bars). The evolutionary history was inferred using the maximum likelihood method based on the Tamura–Nei model. In each case, the tree with the highest log likelihood is shown. Initial trees for the heuristic search were obtained automatically by applying the neighbor-join and BioNJ algorithms to a matrix of pairwise distances estimated using the maximum composite likelihood (MCL) approach and selecting the topology with superior log likelihood values.

**Table 1 animals-15-01636-t001:** Information on the mink studied here and on their trapping points. The coordinates given are those for the Universal Transverse Mercator (UTM) system of the GPS location of the site at which each animal was trapped, for Sector 30 of the Northern Hemisphere of the Earth’s surface. M, male; F, female. Bold type has been used to identify the animals that tested positive for SARS-CoV-2.

Animal	ID	Trapping Date	Riverbed	Belongs to	Coordinates		Weight (g)	Sex
					E:	N:		
1	4618	17 November 2020	Palancia	Navajas	712,550	4,417,756	630	F
2	4619	17 November 2020	Mijares	Onda	737,126	4,429,832	944	M
3	4620	17 November 2020	Mijares	Toga	725,070	4,435,212	771	F
4	4621	17 November 2020	Palancia	Segorbe	717,603	4,412,528	1010	M
5	4622	17 November 2020	Palancia	Soneja	720,334	4,411,080	1109	F
6	4623	17 November 2020	Palancia	Soneja	719,168	4,411,224	1412	M
7	4624	17 November 2020	Palancia	Soneja	721,249	4,410,329	700	F
8	4625	17 November 2020	Palancia	Vall de Almonacid	717,559	4,420,103	590	F
9	4626	17 November 2020	Palancia	Segorbe	716,595	4,413,536	989	M
10	4627	17 November 2020	Palancia	Segorbe	716,102	4,414,547	926	M
11	4628	17 November 2020	Palancia	Segorbe	716,595	4,413,436	1098	M
12	4629	17 November 2020	Palancia	Segorbe	716,102	4,414,547	787	F
13	4630	17 November 2020	Palancia	Jérica	709,496	4,419,036	1083	M
14	4631	17 November 2020	Palancia	Navajas	713,317	4,417,348	1201	M
15	4632	17 November 2020	Palancia	Navajas	712,550	4,417,756	877	M
**16**	**4633**	**17 November 2020**	**Palancia**	**Segorbe**	**717,603**	**4,412,528**	**757**	**F**
17	4634	17 November 2020	Palancia	Jérica	709,496	4,419,036	1179	F
18	4635	17 November 2020	Palancia	Navajas	712,550	4,417,756	993	M
19	4636	17 November 2020	Palancia	Navajas	712,550	4,417,756	493	F
20	4637	17 November 2020	Palancia	Soneja	720,334	4,411,080	646	F
21	4638	17 November 2020	Palancia	Jérica	709,496	4,419,036	1089	M
22	4639	17 November 2020	Palancia	Segorbe	716,595	4,413,436	1074	M
23	4640	17 November 2020	Palancia	Jérica	708,547	4,419,355	1040	M
24	4641	17 November 2020	Palancia	Segorbe	717,603	4,412,528	680	F
25	4642	17 November 2020	Palancia	Soneja	719,168	4,411,224	728	M
26	4643	17 November 2020	Palancia	Viver	705,722	4,420,834	598	F
27	4644	17 November 2020	Palancia	Jérica	706,960	4,421,281	592	F
28	4645	17 November 2020	Palancia	Segorbe	717,603	4,412,528	996	M
**29**	**4646**	**17 November 2020**	**Mijares**	**Vallat**	**727,495**	**4,434,312**	**736**	**F**
30	4647	17 November 2020	Palancia	Jérica	709,496	4,419,036	988	M
31	4648	17 November 2020	Palancia	Viver	705,122	4,420,834	999	M
32	4649	17 November 2020	Palancia	Jérica	706,960	4,421,281	639	F
33	4650	17 November 2020	Mijares	Espadilla	726,089	4,434,535	847	F
34	323	11 March 2021	Palancia	Navajas	712,550	4,417,756	916	M
35	324	11 March 2021	Palancia	Soneja	720,334	4,411,080	1073	M
36	325	11 March 2021	Palancia	Segorbe	716,102	4,414,547	1305	M
37	326	11 March 2021	Palancia	Jérica	709,496	4,419,036	1432	M
38	327	11 March 2021	Palancia	Soneja	720,334	4,411,080	1418	M
39	328	11 March 2021	Palancia	Navajas	713,317	4,417,348	633	F
**40**	**329**	**11 March 2021**	**Palancia**	**Soneja**	**720,334**	**4,411,080**	**1102**	**M**
41	330	11 March 2021	Palancia	Navajas	712,550	4,417,756	777	F
42	331	11 March 2021	Palancia	Viver	704,779	4,419,830	1229	M
43	332	11 March 2021	Palancia	Segorbe	716.595	4,413,436	873	M
44	333	11 March 2021	Mijares	Toga	724,069	4,435,762	1311	M
45	334	11 March 2021	Palancia	Soneja	719,168	4,411,224	1164	M
46	335	11 March 2021	Palancia	Soneja	720,334	4,411,080	1340	M
47	336	11 March 2021	Palancia	Jérica	706,960	4,421,281	680	F
48	337	11 March 2021	Palancia	Navajas	713,317	4,417,348	1311	M
49 ^a^	1000	3 May 2022	Palancia	Gaibiel	715,043 ^b^	4,423,390 ^b^	985	F
50	1007	3 May 2022	Palancia	Jérica	706,960	4,421,281	765	F
51	1008	3 May 2022	Palancia	Segorbe	717,603	4,412,528	867	F
52 ^a^	1009	3 May 2022	Palancia	Jérica	706,960	4,421,281	830	F
53	1010	3 May 2022	Palancia	Jérica	709,496	4,419,036	1421	M
54 ^a^	1013	3 May 2022	Palancia	Jérica	709,496	4,419,036	897	F
55	1014	3 May 2022	Palancia	Jérica	709,496	4,419,036	1389	M
**56**	**1022**	**3 May 2022**	**Palancia**	**Teresa**	**700,530**	**4,419,107**	**702**	**F**
57	1023	3 May 2022	Mijares	Montanejos	712,777	4,439,177	773	F
58	1024	3 May 2022	Mijares	Toga	724,069	4,435,762	1266	M
59	1025	3 May 2022	Palancia	Segorbe	714,680	4,416,113	718	F
60	1026	3 May 2022	Mijares	Toga	725,070	4,435,212	711	F

^a^ Pregnant females. ^b^ Exact location not recorded. The coordinates given are those of the Gaibiel village (200 inhabitants), which is connected by a ravine to the nearby (<4 Km) Regacho water dam of the Palancia river. Bold in the table highlights positive animals.

**Table 2 animals-15-01636-t002:** Animals and samples that were positive for SARS-CoV-2 by qPCR and changes identified in the sequenced regions.

Positive Animal ^a^	Sample	qPCR Results as Ct Values Change in the Coding Sequence of the Gene Change in the Amino Acid Sequence of the Protein ^b^		
		*S* ^c^	*N*	*ORF10*
16	Swab (nasal)	31.3	32.1 *N*:659C > T N: A220V	29.1 *ORF10*: 88G > T ORF10: V30L
29	Swab (rectal)	29.4	32.3 *N*: 608–610 GGG > AAC N: R203K/G204R	20.2 *ORF10*: 88G > T ORF10: V30L
	Lymph node (mediastinal)	30.9	32.1 *N*: 608–610 GGG > AAC N: R203K/G204R	22.8 *ORF10*: 88G > T ORF10: V30L
	Lung tissue	30.9	32.1 *N*: 608–610 GGG > AAC N: R203K/G204R	21.3 *ORF10*: 88G > T ORF10: V30L
40	Swab (nasal)	32.9	32.7 *N*:659C > T N: A220V	29.3 *ORF10*: 88G > T ORF10: V30L
56	Swab (rectal)	29.0	34.4 *N*: 608–610 GGG > AAC N: R203K/G204R	24.9 *ORF10*: 88G > T ORF10: V30L

^a^ Animal identification number as in Table 1. ^b^ Expressed as amino acid changes in the encoded N, S, and ORF10 proteins, relative to the sequences given in GenBank NC_045512.2. ^c^ No mutations identified in the *S* amplicon.

## Data Availability

The data generated and analyzed during this study have been deposited in GenBank. The accession numbers for the sequences are *S* gene region: PQ461249; *N* gene region mutation C28932T: PQ461250; *N* gene region mutation GGG28881-28883AAC: PQ519871; *ORF10* gene region: PQ461251. Additional data supporting the findings of this study are available from the corresponding authors upon reasonable request.

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
