# Peer review of "Four Novel SARS-CoV-2 Infected Feral American Mink (Neovison Vison) Among 60 Individuals Caught in the Wild"

_animals, 2025, doi:10.3390/ani15111636_

Round 1
Reviewer 1 Report
Comments and Suggestions for Authors
The paper details SARS-CoV-2 testing of multiple sample types from introduced feral mink in Spain. The finding of infected individuals and difference in test positivity by sample type provide important information – both regarding tissue tropism and progression of infections in animals as well as the potential for establishment of wildlife reservoirs. A major limitation of the study is the limited sequencing data. Even with high CT values whole genomes or at least targeted open reading frames are possible using a tiled amplicon approach. The addition of this data would greatly improve the interpretations and ability to understand the current state of SARS-CoV-2 infections in feral mink in Spain.
Line 46: Unclear how this is a One Health strategy – it sounds more like animal surveillance. One Health strategies include the environment as well.
Line 53: I suggest “likely” instead of “possibly” given evidence for a wild animal reservoir. The debate centers more around where the initial spillover occurred.
Line 56: Again, the One Health Concept includes the environment as well as animals – this should be made clear and can bring in the encroachments of humans into wildlands as one driver of disease emergence.
Line 74: The serological and PCR data suggests a much higher number of infections in white-tailed deer in North America (~30% infected, so millions of deer). I agree mustelids are important but good to note the deer as well. Also, the farmed nature of mustelids and the strong evidence of multiple spillbacks into humans does arguably make them a bigger concern.
Line 78: I suggest editing “back infection” to “transmission back to humans”
Line 114: Given the importance of sampling to interpreting the results please include details on storage and swabs again here (other capture details are not needed). I also got confused if the media was only for the fetuses or all samples? Please edit this section for clarity.
Table 1 is great but can be put in the supplemental materials based on the editors choice. It is not needed to understand the results, especially given figure 1.
Line 126: I am unclear what is meant by “home devised” I think it means something they came up with, if so I think “custom” is a better word choice here.
Line 130: I am ok with the use of a single gene region for the first screen but compared to many studies this is a small number of samples. Especially given positives were confirmed. I would just delete the explanation – my guess is this is a response to another reviewer but no I feel no need to include it (the fun of peer review).
Line 189 -217: The discussion of the mink weights is all rather interesting, but it seems to not fit the overarching focus of the paper. I wanted to highlight this while also saying I think that this is the best place to include these findings. The writing could be condensed here, and it would be good to understand the authors rationale for including this information.
Line 234: I suggest editing “utilizing for fluorescent detection SYBR green.” To “using SYBR green”
Line 280: Delete “Ones”
Line 293: Viral load does not always directly relate to CT value.
Line 315: With small sample sizes a few individuals can make a large difference in prevalence, so I would not say the lack of significance is overall that surprising.
Line 375: Expanded sequencing would be beneficial and help determine if virus is persisting within Mink. The similarity to the WU1 here suggest it may be but as noted it is very small bits of sequencing.
Line 398- 401: The assumption they were infected by the same variant is not strongly supported by the short sequence reads obtained here.
Line 467-469: I agree that waste water is not the likely source of spread but I think this is a bit of a leap given your data. Can you provide additional rationale here? Are there any studies that do indicate waste water spread to wildlife?
Line 474 – 476: Continued circulation within the mink (or unsampled wildlife) would also explain this result.
Author Response
We thank the reviewer for his/her highly constructive criticisms and careful consideration of the manuscript.
Comments and Suggestions for Authors
Query 1. A major limitation of the study is the limited sequencing data. Even with high CT values whole genomes or at least targeted open reading frames are possible using a tiled amplicon approach. The addition of this data would greatly improve the interpretations and ability to understand the current state of SARS-CoV-2 infections in feral mink in Spain.
Answer. We agree with the reviewer that more extensive sequencing would have been desirable. This is now explicitly mentioned as a limitation in the final paragraph of the Results & Discussion (lines 480-492 of the clean revised version), briefly hinting why it could not be done. We appreciate the suggestion of the reviewer of using a tiled amplicon approach but this technique was not available to us, we have not experience on it, and it appears that it is not free of problems, particularly for untrained users as ourselves. Even with the limitation imposed by the limited sequencing data, the paper conveys a lot of novel and important information, providing insight on SARS-CoV-2 infections in feral mink and conveying the very important good news of low prevalence of the infection among feral mink. Therefore, in the interest of not delaying the availability to the scientific and public health communities of the present data and conclusions, we ask the reviewer to not consider essential a more complete sequencing effort for which success would not be guaranteed, and which would delay much publication of the present data.
Query 2. Line 46: Unclear how this is a One Health strategy – it sounds more like animal surveillance. One Health strategies include the environment as well.
Answer: The reviewer is right. In line with this, and prompted by another comment of Reviewer 2 , we have modified the end of the Abstract (Lines 44-47 of the clean revised version)to read as follows: “Our findings reveal that SARS-CoV-2 circulation is limited in feral mink, at least in this region. They underscore the key importance of wildlife surveillance as an element of the One Health strategy, which encompasses humans, animals and the environment.” We hope that this will be considered appropriate.
Query 3. Line 53: I suggest “likely” instead of “possibly” given evidence for a wild animal reservoir. The debate centers more around where the initial spillover occurred.
Answer. Thanks for the suggestion. Done (line 54 of the clean revised version)
Query 4. Line 56: Again, the One Health Concept includes the environment as well as animals – this should be made clear and can bring in the encroachments of humans into wildlands as one driver of disease emergence.
Answer. The reviewer is right. We have modified the sentence along these lines as follows: “It provided very strong arguments for the One Health concept, which includes animals as well as the environment and the encroachments of humans into wildlands in disease emergence and the development of infectious threats to public health [5].” (lines 56-59 of the clean revised version)
Query 5. Line 74: The serological and PCR data suggests a much higher number of infections in white-tailed deer in North America (~30% infected, so millions of deer). I agree mustelids are important but good to note the deer as well. Also, the farmed nature of mustelids and the strong evidence of multiple spillbacks into humans does arguably make them a bigger concern.
Answer. We thank the reviewer for reminding us of the high number of infected white tail deer in North America. We have reformulated the sentence as follows: “After humans and white tailed deer of North America [12]……” (line 73 of the clean revised version)
Query 6. Line 78: I suggest editing “back infection” to “transmission back to humans”
Answer. Done. We thank the reviewer. (line 79 of the clean revised version)
Query 7. Line 114: Given the importance of sampling to interpreting the results please include details on storage and swabs again here (other capture details are not needed). I also got confused if the media was only for the fetuses or all samples? Please edit this section for clarity.
Answer: This part has been modified by the following addition and/or modification: “In short, at necropsy, each sample was taken and placed aseptically in a plastic tube containing 0.5 ml Sample Preservation Solution (ref. P042T0020100, JiangSu Mole Bioscience, Taizhou, China; sold in Spain by Palex Medical, Madrid, Spain). This proprietary commercial solution is used to inactivate the virus and to preserve the RNA. Animals 49, 52 and 54 (Table 1) were pregnant females. One fetus from each pregnant animal was taken randomly among the several gestation sacs of the pregnancy (see below Figure 3G), for viral assay. The fetus was placed in its tube with 1.5 ml of Sample Preservation solution, and was otherwise treated as the lung and mediastinal lymph node samples. The hermetically sealed tubes containing the samples in preservation solution were placed at −80°C <2 h after procurement, remaining at this temperature until their use in SARS-CoV-2 analyses.” (lines 118-128 of the clean revised version)
Query 8: Table 1 is great but can be put in the supplemental materials based on the editors choice. It is not needed to understand the results, especially given figure 1.
Answer. We are sorry to disagree with the reviewer on this point. We think that keeping the Table within the text, at or near its present location, is of much help to the reader, since reference to this table is frequent throughout the text, and particularly in the initial parts of the Results and Discussion. Thus, it would be cumbersome to move continuously to and from the supplementary material as the reading proceeds. It is true that some information from the Table is conveyed in panels B and C of Fig. 1. However, these panels are designed to provide visual locational information concerning distances between trapping sites, but they convey less information concerning animals, and no information concerning weights. Therefore, if at all possible, we would very much like to retain Table 1 at its present location. However, if considered essential, we would be ready to transfer this Table to the supplementary material.
Query 9. Line 126: I am unclear what is meant by “home devised” I think it means something they came up with, if so I think “custom” is a better word choice here.
Answer: We thank the reviewer for this suggestion. We have replaced “home” by “custom”. (line 135 of the clean revised version)
Query 10. Line 130: I am ok with the use of a single gene region for the first screen but compared to many studies this is a small number of samples. Especially given positives were confirmed. I would just delete the explanation – my guess is this is a response to another reviewer but no I feel no need to include it (the fun of peer review).
Answer. We appreciate this suggestion. We have removed the mention to the use of large number of samples, and reordered the sentences for compactness, as follows: “Initial detection was based on qPCR amplification of nucleotides 28,701-28,951 of the viral genome (from here on, the numbering is that for the Wuhan-1 viral genome sequence, GenBank NC_045512.2) using primers 5’GCAGTCAAGCCTCTTCTCGT3′ and 5’TTGCTCTCAAGCTGGTTCAA3'. This amplicon corresponds to a highly conserved region of the nucleocapsid (N) gene of the virus [20].” (lines 139-144 of the clean revised version).
Query 11. Lines 189 -217: The discussion of the mink weights is all rather interesting, but it seems to not fit the overarching focus of the paper. I wanted to highlight this while also saying I think that this is the best place to include these findings. The writing could be condensed here, and it would be good to understand the authors rationale for including this information.
Answer: We thank the reviewer for appreciating this section. We felt that the data should be included for the sake of initial characterization of the life cycle in this endemic localism. For brevity, we have added towards the beginning of this section (lines 190-191 of the clean revised version) “Although tangential to this study…” to introduce that we were using weights to try to get some insight on the population. In addition to this we have compacted the text as much as we could, although the level of shortening is modest to keep the meaning and level of explanation provided.
Query 12. Line 234: I suggest editing “utilizing for fluorescent detection SYBR green.” To “using SYBR green”
Answer. Done as suggested (line 247 of the clean revised version)
Query 13. Line 280: Delete “Ones”
Answer. Done as requested (line 293 of the clean revised version)
Query 14. Line 293: Viral load does not always directly relate to CT value.
Answer. The reviewer is right. Therefore we have changed the sentence to: “Overall, the results are indicative of low prevalence of the virus among these 60 feral animals and suggest low viral load in those animals that hosted the virus.” (line 308 of the clean revised version)
Query 15. Line 315: With small sample sizes a few individuals can make a large difference in prevalence, so I would not say the lack of significance is overall that surprising.
Answer. We agree with this comment. To align with it, we have condensed the sentence as follows: “The proportion of the trapped animals that were found to be infected was 6.8% for the Palancia river and 14.3% for the Mijares river, two values that were not statistically different (Fisher’s exact test).” (lines 328-330 of the clean revised version)
Query 16. Line 375: Expanded sequencing would be beneficial and help determine if virus is persisting within Mink. The similarity to the WU1 here suggest it may be but as noted it is very small bits of sequencing.
Answer. The reviewer is right, but see our reply to Query 1. Lacking expanded sequencing, we prefer not to elaborate further, to minimize speculation. Thus, this part of the text has been kept as it was.
Query 17. Line 398- 401: The assumption they were infected by the same variant is not strongly supported by the short sequence reads obtained here.
Answer. The reviewer is right that, because of the short sequence, our data do not prove that the two animals were infected by the same variant. However, this possibility is at least tenable on the basis of the limited sequencing data and by the geographic nearness of the trapping sites. Wer agree that a less assertive statement should be made. We hope that the following formulation, used now in the revised text, will be found appropriate: “Animals 16 and 40 were trapped in geographically close places in the same river (Palancia), and thus it would not be surprising if they were infected with the same viral variant, which might have persisted in these locations at the two times of capture (mid-November 2020 and mid-March 2021).” (lines 415-418 of the clean revised version)
Query 18. Line 467-469: I agree that waste water is not the likely source of spread but I think this is a bit of a leap given your data. Can you provide additional rationale here? Are there any studies that do indicate waste water spread to wildlife?
Answer. We really cannot provide positive published information that proves waste water spread to wildlife. However, our findings in our previous pilot paper on mink (reference 7) and on an infected otter (reference 8), and those presented here would fit wastewater as a disease-transmitting matrix, although with not high efficiency. We agree that our formulation conveys the impression of a leap. To alleviate this impression, and also to cope with a suggestion of reviewer 2, we have reformulated the last paragraph of the Results & Discussion section to read as follows: “Our study is not exempt from limitations. The more important one concerns our inability to draw further conclusions about potential variants due to our limited sequencing approach. The fact that we could not carry out next-generation sequencing (NGS) on our samples was largely due to the high Ct values, which were on average above 30 and that we equated with a low viral load, while NGS typically requires a high viral load for reliable sequencing. Despite this limitation, our phylogenetic analysis provided valuable insights concerning variants, while the combination of the observed mutations and low viral load supports our suggestion that the life cycle of SARS-CoV-2 transmission among feral mink is not of major concern at this time. Nevertheless, this conclusion should not result in halting the monitoring and the genetic sequencing even in the limited dimension done here, since these measures are essential to assess potential wildlife adaptation of the virus that could result in novel zoonotic risks in case of re-entry of new variants into the human population.” (lines 480-492 of the clean revised version)
Query 19. Line 474 – 476: Continued circulation within the mink (or unsampled wildlife) would also explain this result.
Answer. We agree with this alternative possibility. However, to avoid being too speculative, we have decided to remove this sentence.
Reviewer 2 Report
Comments and Suggestions for Authors
-Abstract
Key message buried: The important conclusion about "limited circulation" of SARS-CoV-2 is not emphasized upfront.
- Introduction
Weak transition from previous work: The bridge from the pilot study to the current expanded study lacks logical flow.
- Materials and Methods
Missing summarized tables: For better readability, a concise table summarizing sample types and numbers (especially fetuses) would improve clarity.
- Results
Poor separation of findings: Different aspects (e.g., trapping data, weight analysis, infection status) are not neatly separated, impairing readability.
- Discussion
Logical jumps: Connections between environmental contamination, fecal-oral transmission, and low infectivity are somewhat speculative and not logically tight.
Insufficient emphasis on limitations: Limitations (e.g., inability to perform NGS due to high Ct values) are mentioned but should be more explicitly acknowledged, preferably in a distinct "Limitations" paragraph.
Author Response
We thank the reviewer for his/her highly constructive comments
Query 1. -Abstract. Key message buried: The important conclusion about "limited circulation" of SARS-CoV-2 is not emphasized upfront.
Answer. Thanks for this comment. We have tried to address it by modifying the penultimate sentence of the Abstract to: “Our findings reveal that SARS-CoV-2 circulation is limited in feral mink, at least in this region.” (lines 44-47 of the clean revised version)
Query 2. -Introduction. Weak transition from previous work: The bridge from the pilot study to the current expanded study lacks logical flow.
Answer. We apologize for not having been more explicit. The finding of two cases among 13 animals in the pilot study was worrysome to us, given the low number of individuals in that initial study, so it was evident to us that a larger sample had to be examined. We hope to have translated this to the reader by modifying a sentence towards the final part of the Introduction as follows: “With the purpose of getting sounder indications on how frequent was SARS-CoV-2 infection of feral mink in these two river courses, we extended the study to sixty additional feral animals culled in these river courses in three campaigns taking place from November 2020 till May 2022.” (lines 89-93 of the clean revised version)
Query 3. -Materials and Methods. Missing summarized tables: For better readability, a concise table summarizing sample types and numbers (especially fetuses) would improve clarity.
Answer. All animals were identical concerning the samples taken and analyzed (except the three pregnant females, that provided, in addition, one fetus). Thus, instead of adding a Table, to comply with the request of the reviewer, we have added procedural detail in the Materials and Methods section as follows: “The treatment of the animals, including humane sacrifice, conservation, examination, necropsy and collection of samples (presently nasal and rectal swabs, mediastinal lymph nodes and lung tissue from all 60 animals) was reported in our previous works [7,8]. In short, at necropsy, each sample was taken and placed aseptically in a plastic tube containing 0.5 ml Sample Preservation Solution (ref. P042T0020100, JiangSu Mole Biosci-ence, Taizhou, China; sold in Spain by Palex Medical, Madrid, Spain). This proprietary commercial solution is used to inactivate the virus and to preserve the RNA. Animals 49, 52 and 54 (Table 1) were pregnant females. One fetus from each pregnant animal was taken randomly among the several gestation sacs of the pregnancy (see below Figure 3G), for viral assay. The fetus was placed in its tube with 1.5 ml of Sample Preservation solution, and was otherwise treated as the lung and mediastinal lymph node samples. The hermetically sealed tubes containing the samples in preservation solution were placed at −80°C <2 h after procurement, remaining at this temperature until their use in SARS-CoV-2 analyses.” (lines 116-128 of the clean revised version). We hope that this solution properly addresses this concern of the reviewer.
Query 4.- Results. Poor separation of findings: Different aspects (e.g., trapping data, weight analysis, infection status) are not neatly separated, impairing readability.
Answer. We apologize for the lack of neat separation of these parts. To help the separation and the readibility, and also to comply with a query of Reviewer 1, we have:
- a) Modified the writing of the first paragraph of the second section of the Results & Discussion (the one on the weights), compacting it and mentioning that it is tangential to the paper but important for feral population characteristics (lines 189-192 of the clean revised version).
- b) Added to the third section of the Results & Discussion (the one on the infection status) an introductory sentence stating the goal, also modifying the ensuing two sentences to make them more direct and to connect them with the remainder of the paper. This part now reads: “The major aim of our study was to assess the infection status of the trapped animals concerning SARS-CoV-2. With this goal, we carried out qPCR studies on the cDNA obtained by retrotranscription of total RNA isolated from four samples (nasal and rectal swabs and lung and mediastinal lymph node tissues) obtained from each one of the 60 animals. As already indicated in the Materials and Methods, and with the same purpose of SARS-CoV-2 detection, we also obtained RNA from a fetus taken randomly from each one of the three pregnancies found at necropsy in female animals 49, 52 and 54 (Table 1).” (lines 238-244 of the clean revised version)
- c) In addition, we have modified the second paragraph of this same section (the third section of the Results & Discussion; on the infection status), particularly its second half, to make it clearer, shorter and more direct, and also to address a query by reviewer 1. This second part of the paragraph now reads: “The two animals (animals 16 and 40) for which the nasal swab was positive, exhibited qPCR positivity with Ct values ≥29 for all three genetic regions examined, suggesting low viral loads. Animal 29, the one for which lung and mediastinal lymph node were positive, presented the lowest Ct values among the four SARS-CoV-2 positive animals, suggesting that it presented the highest viral load. Even in this case, the Ct values were >20 in the qPCR assays for the three genes in the three positive swab/tissue samples, supporting that the viral load was not very high.” (lines 300-306 of the clean revised version).
Query 5. -Discussion. Logical jumps: Connections between environmental contamination, fecal-oral transmission, and low infectivity are somewhat speculative and not logically tight.
Answer. To cope with this query we have reordered more logically the sentences in this part of the manuscript, also shortening them and eliminating undue speculation. The shortenerd text now reads as follows: “We previously hypothesized for mink [7] and for wild otter [8] that SARS-CoV-2-contaminated waters might be the original source of the virus in those animals that are infected. SARS-CoV-2 has been shown to persist for some time in water [15] and to be present in human stools [30] where it appears to derive largely from human excreta [15, 31,32]. Specifically, in the province where our mink live (Castellón), Barberá-Riera et al. [29] detected SARS-CoV-2 in wastewater and recovered the genome sequence of the dominant viral variant in humans at the time of their study, (B.1.177), whose sequence is similar to the one in some of our trapped feral mink, as shown by our phylogenetic analyses. Nevertheless, the low frequency of infection among our animals suggests that the waters of these two rivers do not sustain high transmission rates.” (lines 455-464 of the clean revised version)
Query 6. -Discussion. Insufficient emphasis on limitations: Limitations (e.g., inability to perform NGS due to high Ct values) are mentioned but should be more explicitly acknowledged, preferably in a distinct "Limitations" paragraph.
Answer. We have tried to comply with this suggestion of the reviewer by modifying the last paragraph before the Conclusions, also taking in consideration a query by Reviewer 1. The paragraph now reads as follows: “Our study is not exempt from limitations. The more important one concerns our inability to draw further conclusions about potential variants due to our limited sequencing approach. The fact that we could not carry out next-generation sequencing (NGS) on our samples was largely due to the high Ct values, which were on average above 30 and that we equated with a low viral load, while NGS typically requires a high viral load for reliable sequencing. Despite this limitation, our phylogenetic analysis provided valuable insights concerning variants, while the combination of the observed mutations and low viral load supports our suggestion that the life cycle of SARS-CoV-2 transmission among feral mink is not of major concern at this time. Nevertheless, this conclusion should not result in halting the monitoring and the genetic sequencing even in the limited dimension done here, since these measures are essential to assess potential wildlife adaptation of the virus that could result in novel zoonotic risks in case of re-entry of new variants into the human population.” (lines 480-492 of the clean revised version).
Round 2
Reviewer 2 Report
Comments and Suggestions for Authors
You've answered and corrected most of the points I pointed out well. Thanks for your hard work.